# Serrated Colorectal Lesions: An Up-to-Date Review from Histological Pattern to Molecular Pathogenesis

**DOI:** 10.3390/ijms23084461

**Published:** 2022-04-18

**Authors:** Martino Mezzapesa, Giuseppe Losurdo, Francesca Celiberto, Salvatore Rizzi, Antonio d’Amati, Domenico Piscitelli, Enzo Ierardi, Alfredo Di Leo

**Affiliations:** 1Section of Gastroenterology, Department of Emergency and Organ Transplantation, University of Bari, 70124 Bari, Italy; martino.mezzapesa@gmail.com (M.M.); giuseppelos@alice.it (G.L.); celibertofrancesca@gmail.com (F.C.); salvatore.rizzi.med@gmail.com (S.R.); alfredo.dileo@uniba.it (A.D.L.); 2PhD Course in Organs and Tissues Transplantation and Cellular Therapies, Department of Emergency and Organ Transplantation, University of Bari, 70124 Bari, Italy; 3Section of Pathology, Department of Emergency and Organ Transplantation, University of Bari, 70124 Bari, Italy; antonio.damati@uniba.it (A.d.); domenico.piscitelli@uniba.it (D.P.)

**Keywords:** colorectal cancer, colorectal serrated lesions, sessile serrated lesions, serrated pathway, microsatellite instability, CpG island methylator phenotype, molecular pathways

## Abstract

Until 2010, colorectal serrated lesions were generally considered as harmless lesions and reported as hyperplastic polyps (HPs) by pathologists and gastroenterologists. However, recent evidence showed that they may bear the potential to develop into colorectal carcinoma (CRC). Therefore, the World Health Organization (WHO) classification has identified four categories of serrated lesions: hyperplastic polyps (HPs), sessile serrated lesions (SSLs), traditional serrated adenoma (TSAs) and unclassified serrated adenomas. SSLs with dysplasia and TSAs are the most common precursors of CRC. CRCs arising from serrated lesions originate via two different molecular pathways, namely sporadic microsatellite instability (MSI) and the CpG island methylator phenotype (CIMP), the latter being considered as the major mechanism that drives the serrated pathway towards CRC. Unlike CRCs arising through the adenoma–carcinoma pathway, APC-inactivating mutations are rarely shown in the serrated neoplasia pathway.

## 1. Introduction

Colorectal cancer (CRC) is a major public healthcare problem, associated with a high rate of morbidity and mortality in the Western area and in other countries with comparable lifestyle and dietary habits. Indeed, CRC represents the second leading cause of cancer-related death in the United States and the fourth leading cause of cancer-related death worldwide [1].

CRC represents the common endpoint of a wide range of genetic (gene mutations, chromosomal abnormalities) and epigenetic changes occurring in normal colorectal cells. As a result, normal colorectal epithelium evolves towards precancerous lesions that lead to the development of CRC. Approximately 85% of CRCs have been suggested to develop from the malignant transformation of benign adenomas and colorectal polyps [2]. Histologically, colorectal polyps may have the characteristics of conventional adenomas (tubular, tubulovillous and villous), colorectal serrated lesions and other polyps, such as juvenile and inflammatory ones, with extreme variability in their carcinogenic risk. Conventional adenomatous polyps and sessile serrated lesions (SSLs) are recognized as the most likely precursors of CRCs. Since conventional adenomas and SSLs are estimated to be present in 20% to 53% of subjects older than 50 years of age, [3,4,5,6,7,8] they have a challenging impact on healthcare.

Although a lot of studies have clearly described the mechanisms of adenoma carcinoma sequence, less evidence has been retrieved about serrated polyp-related CRC. Therefore, the aim of the present paper is to depict a scenario of colorectal serrated lesions, which are precursors of approximately one-third of colorectal cancers. In detail, we aim to provide an up-to-date review of the molecular features that cause transformation of normal epithelium to colorectal serrated lesions and from colorectal serrated lesions to CRC. 

## 2. The WHO Classification of Colorectal Serrated Lesions

Until 2010, colorectal serrated lesions were generally considered as harmless lesions and reported as hyperplastic polyps (HPs) by pathologists and gastroenterologists [9]. Thereafter, a multitude of terms were coined to describe these lesions, making a strong contribution to misunderstandings about serrated lesion terminology and classification. On these bases, a rational classification from a histopathological point of view was provided in 2010 by the World Health Organization (WHO) [10], as shown in Table 1.

Colorectal serrated lesions should nowadays be classified according to the WHO classification of tumors of the digestive system, which was updated in 2019 to the 5th edition [11]. This new classification differentiates serrated lesions of the colorectum into four general categories (as shown in Table 2).

Hyperplastic polyps (HPs)Sessile serrated lesions (SSLs)Traditional serrated adenoma (TSAs)Serrated adenomas, unclassified

The major edit in terminology is the approval of the new term “sessile serrated lesions” (SSLs), which aims to overcome the previous confounding terminologies “sessile serrated adenoma/sessile serrated polyp”. Indeed, the term “adenoma” encompasses the concept of dysplasia, which is not shown in a considerable number of SSLs. Moreover, some SSLs may not show a polypoid appearance, which makes the term “polyp” unsuitable for the purpose. The term “lesion” replaces the plethora of terms used in the past and appears both biologically and pathologically accurate, since it does not imply the presence of dysplasia [12]. Indeed, the finding of a dysplastic pattern in SSLs requires the specific diagnostic terminology of “SSL with dysplasia”.

Minor changes in the new classification include the elimination of mucin-poor HP (MPHP) among the HP subtypes [11] and the introduction of a new entity “serrated adenoma, unclassified”. The removal of MPHP is justified by an unproven clinical significance of HP subtyping [13]. “Serrated adenoma, unclassified” refers to those ambiguous colorectal polyps showing both dysplasia and serrated architecture, which cannot be clearly classified as SSL, TSA, or conventional adenoma. 

## 3. Colorectal Serrated Lesions: Histologic and Endoscopic Features

Microscopically, colorectal serrated lesions are defined by the presence of sawtooth-like in-folding of the epithelial crypts [11]. The differences between the main subtypes of serrated lesions dwell in cytological/architectural features and location/extent of proliferative zone [11,14,15]. These differences are supposed to result from different genetic and epigenetic alterations in genes responsible for cell proliferation and differentiation.

### 3.1. Hyperplastic Polyps (HP)

HPs are the most common serrated lesions, as they account for approximately 75% of all serrated polyps [16] with an estimated prevalence in patients undergoing screening colonoscopies of 20–30% [17,18,19,20,21,22]. In HPs, crypts are straight and elongated, with a serrated architecture confined to the upper two-thirds of the crypts, and cells show only minimal cytological atypia, as shown in Figure 1. On the other hand, in the basal third of the crypt, architecture is regular and non-serrated, and cells do not show any sign of atypia. Two variants of HPs have been identified: microvescicular hyperplastic polyps (MVHPs) and goblet-cell-rich hyperplastic polyps (GCHPs). GCHPs show, as suggested by their name, an increased number of goblet cells forming small polyps, whereas MVHPs are recognized by the presence of small droplets (microvescicular) of mucin within the cytoplasm of most cells. MVHPs can be more likely found in the left colon, just like CGHPs which are typically very small (<0.5 cm). MVHPs are considered precursors of SSLs (see below). During colonoscopies, HPs are commonly found in the left colorectum, (most of them in the rectosigmoid colon), are often ≤5 mm in size and in white light endoscopy, they appear as roundish pale flat-elevated or sessile lesions, occasionally covered with normal mucosa. In chromoendoscopy, Kudo type II asteroid pits are observed on the lesion surface [16,17,18,19,20,21,22].

### 3.2. Sessile Serrated Lesions (SSLs)

SSLs represent the second most common type of serrated lesion and, along with TSAs, are considered significant precursor lesions of CRC. SSLs (previously known as SSA/SSP) have a significantly lower prevalence (when compared with HPs) in the general population which amounts to about 5–10% [16,17,18,19,20,21,22]. SSLs are generally larger than HPs, as their average diameter usually reaches 5–7 mm, and their shape is flat or sessile rather than polypoid. Unlike HPs, SSLs can be more frequently discovered in the right colon [16]. The major and pathognomonic histologic feature that distinguishes SSLs from HPs is the finding of architecturally distorted serrated crypts, which is the result of an overlap of alterations in the proliferative zone of the crypts. In order to be considered diagnostic for SSL a crypt should show at least one of the following histologic features: (1) Horizontal growth along the muscularis mucosa (L-shaped or inverted T-shaped crypt); (2) Dilation of the crypt base (basal one-third of the crypt); (3) Serrations extending into the crypt base; (4) Asymmetrical proliferation (shift of the proliferation zone from the base to the lateral side). A picture of an SSL is shown in Figure 2. According to the new diagnostic criteria established by the WHO in 2019, the presence of at least one unequivocal “architecturally distorted serrated crypt” is mandatory for diagnosis [11]. These new criteria are significantly clearer than the previous ones, which considered as diagnostic the presence of “two or three” distorted serrated crypts [10]. The endoscopic features of SSLs mainly include the presence of a mucous cap with a cloud-like surface in white light endoscopy [16,23] and type II open-shape (type II-O) pits in chromoendoscopy [24]. SSLs often appear discolored or whitish. Since most of their endoscopic characteristics are similar (shape, size, color) to HPs, endoscopic differential diagnosis between SSLs and HPs remains challenging.

#### Sessile Serrated Lesions with Dysplasia (SSL-Ds)

In harmony with their well-known carcinogenic potential, SSLs can give rise to dysplastic foci (Figure 3). This histologic entity, named SSLs with dysplasia (SSL-D), represents a subgroup of SSLs. It has been estimated that approximately 4–8% of SSLs contain dysplasia [21,25]. At least three different morphologic types of dysplasia have been described in SSL-D [26,27]. There might be an intestinal dysplasia (adenoma-like) which is similar to the dysplasia observed in conventional adenomas (tubular or tubule-villous), which is relatively rare [26,27]. In contrast, serrated dysplasia is more common and is characterized by eosinophilic cytoplasm and tightly packed small glands, and its presence can be considered to represent progression to TSA [28]. The additional pattern is the minimal deviation dysplasia, with limited changes compared with SSL, and characteristic loss of MLH1 [29]. Most SSL-Ds, however, have an undefined pattern of dysplasia [16,26]. These histologic differences have questionable clinical value. Preliminary evidence indicates that a substantial portion of these lesions demonstrates inactivation of the mismatch repair gene MLH1, and the dysplastic areas often demonstrate microsatellite instability (see below). Endoscopically, these lesions are likely to present large or small nodules on the surface (93% of cases) [29] and in chromoendoscopy they exhibit an adenomatous pit pattern (Kudo type III, IV) in 70–95% [30,31].

### 3.3. Traditional Serrated Adenoma (TSA)

Similarly to HPs, TSAs more commonly appear in the left colon. They are the most uncommon serrated lesions of the colorectum, as their prevalence is <1% in the general population undergoing screening colonoscopy [17,18,19,20,21,22]. Usually, TSAs are larger than SSLs and have a polypoid or pedunculated appearance [16]. Histologically, TSAs show a distorted villous (filiform) or tubulovillous architecture, and in many cases the villi have bulbous tips. From a cytological point of view, TSAs have predominantly cells with eosinophilic cytoplasm and a basal/central elongated nucleus (penicillate nuclei), as shown in Figure 4. The epithelium is usually pseudostratified, but cells are usually not in proliferating activity as observed by Ki67 immunoreactivity [14,32,33]. There is no universal agreement about the nature of these lesions, since it has not been fully established whether they represent a kind of “metaplasia” rather than true dysplasia. In TSAs, both conventional adenoma-like dysplasia and serrated dysplasia can be observed. Prior to the development of CRC, it is believed that TSAs acquire increasing degrees of cytological atypia. The risk of malignancy in TSA, and the rapidity of progression to carcinoma, is unknown. As discussed in the molecular section of this article, there is a greater heterogeneity in the molecular profile of TSA compared with SSA/P. This may be partly due to inconsistency in the diagnostic criteria used for this lesion. They appear as reddish, protruded or pedunculated lesions in white light observation and “pinecone-like” or “branch coral-like” lesions in macroscopic observation. In chromoendoscopy, they show type III_H_ pits, which are Kudo type III_L_-like tubular pits, or type IV_H_ pits, which are Kudo type IV-like villous pits [34].

### 3.4. Serrated Polyposis Syndrome (SPS)

SPS is a rare condition, characterized by the presence of multiple serrated colorectal lesions throughout the large bowel, with an increased risk of developing CRC. The prevalence of SPS in the population undergoing colonoscopy ranges from 0.03% to 0.5% [35]. SPS diagnosis is usually reported at the age of 49–56 years, due to colonoscopy screening procedures, although an age range of 22–71 has been observed and the disorder is equally common among men and women [35,36]. WHO recently updated the diagnostic criteria for SPS [11]. WHO defines the syndrome by the presence of any one of the following conditions: (1) at least five serrated lesions or polyps proximal to the rectum >5 mm, with two or more bigger than 10 mm, (2) more than twenty serrated lesions or polyps of any size distributed throughout colorectum, with at least five proximal to the rectum [11]. Former WHO criteria for SPS included (3) any number of serrated polyps proximal to the sigmoid colon in an individual with a first-degree relative with SPS [10]. Any serrated polyp histological subtype (SSLs, HPs, TSAs, serrated adenoma unclassified) is included in the polyp count. The count even includes serrated lesions detected over multiple colonoscopies. The endoscopic and histologic features of serrated lesions that can be found in SPS obviously depend on the histological subtype to which they belong (SSLs, HPs, TSAs). Studies from cohorts of patients affected by SPS report an overall cancer risk in the range of 15–30%. This variability depends on patient age, polyp phenotype and the presence of high-risk histologic features [36,37]. Close endoscopic surveillance (one colonoscopy per year) is recommended [38]. 

## 4. Molecular Pathways Leading from Normal Mucosa to CRC

Most CRCs are sporadic tumors (75%), arising in patients with no family history or apparent genetic predisposition, whereas about 20% of CRC patients report a family history [39,40]. Only 3–5% of CRCs are considered hereditary. Regardless of genetic background, carcinogenesis is a multistage process which involves a sequence of multiple genetic events. Indeed, CRC pathogenesis is due to the sequential acquisition of genetic and epigenetic alterations, which are confirmed to drive the activation of oncogenes or inactivation of oncosuppressor genes. These events allow normal epithelial cells to progressively convert into undifferentiated cells with impaired, autonomous and uncontrolled growth mechanisms, able to evade apoptosis and infiltrate healthy tissues.

### 4.1. Hereditary CRCs

Hereditary CRC develops in patients bearing germline mutations associated with well-defined cancer-predisposing syndromes such as familial adenomatous polyposis (FAP) and Lynch syndrome (previously known as hereditary nonpolyposis colorectal cancer, HNPCC). Patients affected by FAP (<1% cases of CRC), an autosomal dominant disease, inherit a germline mutation on the APC allele and precancerous lesions develop when the second allele acquires further genotoxic damage, becoming mutated or deleted, since APC represents a tumor-suppressor protein acting as an antagonist of the WNT-signaling pathway [41]. Lynch syndrome, the most common form of hereditary CRC (1–3% cases), is characterized by microsatellite instability (MSI) as a consequence of a germline mutation in a DNA mismatch repair (MMR) gene [42,43]. Finally, a mention should be reserved for further rare forms of hereditary CRC, i.e. hamartomatous polyposis syndromes (Peutz-Jeghers syndrome, juvenile polyposis, Cowden disease) and MYH-associated polyposis (MAP) which are globally responsible for <1% CRCs [44].

### 4.2. The Conventional Model of Colorectal Carcinogenesis in Sporadic CRCs

The conventional model of carcinogenesis (Figure 5), the so-called adenoma–carcinoma sequence, begins with normal colorectal epithelium converting in an adenoma (villous, tubulovillous or tubular) and then progresses, through the accumulation of genetic and epigenetic aberrations, to the development of a CRC [45,46,47]. At the molecular level, tumorigenesis proceeds via two major mechanisms: (1) chromosomal instability (CIN) or (2) microsatellite instability (MSI) [41,48,49,50] (Figure 1). In both the CIN pathway and the MSI pathway leading to CRC via the adenoma–carcinoma multistep pathway, dysregulation of the Wnt/β-catenin signaling pathway represents the first step. The *APC* gene on the long arm of chromosome 5 is considered the “gatekeeper” for carcinogenesis. It has been supposed that most adenomas arise from an initial loss of the *APC* gene function, and for that to happen, an epithelial cell must lose the function of both *APC* alleles.

#### 4.2.1. CIN Pathway

CIN represents the most common carcinogenetic pathway in CRC, as it is detected in 85% of sporadic CRCs. In CIN, chromosomes show noteworthy damagesuch as gain or loss of the whole chromosome, or large portions of it [51]. This karyotypic variability can show up either with loss of heterozygosity (LOH) in tumor-suppressor genes or amplification of oncogenes [52,53]. Colorectal carcinogenesis is supposed to proceed according to a universally accepted model, in which bi-allelic mutation in the tumor suppressor gene APC occurs as the first event leading from normal mucosa to early adenoma. Further genetic mutations are needed in order to develop CRC, such as oncogenic K-ras activating mutations in the adenomatous stage, and eventually, deletion of chromosome 18q with inactivation of tumor-suppressor SMAD genes (SMAD2–SMAD4) and inactivation of the tumor-suppressor gene TP53 on chromosome 17p [54,55].

#### 4.2.2. Microsatellite Instability 

MSI is the underlying genetic substrate in 15–20% of sporadic CRC and in more than 95% of Lynch syndrome [48,49]. This pattern of genomic instability is due to the loss of expression of DNA mismatch repair proteins (usually mulL homolog-1 MLH1 or mutS homolog-2 MSH2, but also MSH6, and PMS2) whose function is to correct replication errors of the DNA polymerase [56,57,58]. Impairment of MMR genes can occur in two different ways: sporadic MSI due to an epigenetic mechanism with promoter hypermethylation of the mismatch repair gene MLH1 resulting in the silencing of this gene [6], and the familial form of MSI (Lynch syndrome) in patients bearing germline mutations in the mismatch repair genes MLH1, PMS2, MSH6 or MSH2, which accounts for about 80% of MSI CRCs [59,60]. DNA MMR enzyme deficiency results in an accumulation of errors in DNA sequences, especially in insertions/deletions of short nucleotide repeats within microsatellites. Microsatellite sequences are repetitive sequences of non-coding DNA, usually several (one to five) base pairs in length. In order to define CRC due to MSI, a standard panel of five microsatellite markers should be tested (two mononucleotides, BAT26 and BAT25, and three dinucleotides, D2S123, D5S346 and D17S250) [61]. When inactivation of a DNA mismatch repair gene occurs, these mutations within microsatellites may be detected at different frequencies throughout the genome, hence the terms high MSI (MSI-H) when ≥30% of the markers exhibit instability, low MSI (MSI-L) for those with <30% markers exhibiting instability and stable microsatellite (MSS) when no marker shows instability [62]. Small insertions/deletions may result in frame-shift mutations within repetitive tracts in the coding region of essential tumor-suppressor genes and oncogenes [63]. Most frequently, the genes undergoing mutation are: transforming growth factor-β receptor (TGFBR2), insulin-like growth factor 2 receptor (IGFR-2), BCL2 associated X apoptosis regulator (BAX), human mutS homolog 3 and NADH-ubiquinone oxidoreductase [64]. Even in MSI, the pathway leading to the development of CRC through adenoma–carcinoma sequence presupposes an impairment of Wnt/β-catenin signaling. According to the Cancer Genome Atlas study results, CIN and MSI pathways in carcinogenesis via the adenoma–carcinoma pathway are mutually exclusive [65].

### 4.3. The Serrated Neoplasia Pathway

Since serrated colorectal lesions were recognized as premalignant lesions leading to CRC, in addition to the conventional morphologic adenoma–carcinoma sequence a new morphologic pathway, the so-called “serrated neoplasia pathway”, was described. In past decades, researchers tried to understand the molecular basis of this new pathway able to convert normal mucosa in a serrated colorectal lesion and then in CRC [64,65,66,67,68]. At the molecular level, CRCs arising from serrated lesions originate via two different molecular pathways, namely sporadic MSI and CpG island methylator phenotype (CIMP), the latter being considered as the major mechanism that drives the serrated pathway toward CRC [69]. Unlike CRCs arising through the adenoma–carcinoma pathway, APC-inactivating mutations are rarely shown in the serrated neoplasia pathway. Moreover, when APC mutations are found, they do not act as a first step to carcinogenesis. Most CRCs arising through the serrated neoplasia pathway carry a BRAF mutation, whereas K-ras mutations are less frequent [16].

#### CpG Island Methylator Phenotype (CIMP)

CIMP represents the major genetic substrate of the serrated pathway. Epigenetic regulation of gene expression is a normal mechanism playing a fundamental role in the preservation of genomic stability [70]. CpG (cytosine preceding guanine) islands are regions within the genome commonly located in promoter sites abounding in CpG dinucleotides. Several CpG dinucleotides can be inappropriately unmethylated or aberrantly hypermethylated by DNA methyltransferases (DNMT) causing either hyperactivation or inappropriate silencing of tumor suppressor gene expression [71,72], the latter representing the most common presentation. Interestingly, assessing the methylation status of specific genes in CRC could be useful in clinical practice as it could represent a biomarker for the detection or monitoring of CRC, and more particularly of serrated lesions associated with a CIMP signature [73,74]. CIMP represents a distinct phenotype in CRC, with specific clinical, pathological and molecular features. Since methylation represents a physiological epigenetic regulation, defining CIMP remains challenging. The CIMP status of CRC is usually assessed by a testing CRC for a panel of methylation markers categorizing CRC as exhibiting or not exhibiting DNA methylation based on certain thresholds. Most studies have commonly defined CIMP by using a classic panel containing hMLH1, p16, MINT1, MINT2 and MINT31 [75,76] and IGFBP3, IGF2, RUNX3 and others [77]. Moreover, once hypermethylation of these loci has been identified, studies give different classification to the lesions tested. According to Weisemberg et al. [78] CRCs were classified as CIMP-negative (CIMP-) and CIMP-positive (CIMP+). Ogino et al. tested CRCs for eight CIMP-specific gene promoters and classified CRC in three subgroups: if one to five out of eight markers were hypermethylated they were identified as CIMP-low (CIMP-L), when none of the markers was hypermethylated, CIMP-0, and if six to eight out of eight markers had promoters that were hypermethylated, these were identified as CIMP-high (CIMP-H) [79,80]. As previously reported for the adenoma–carcinoma sequence, molecular pathogenesis proceeds via the CIN pathway and MSI, which are mutually exclusive. In the serrated pathway the CIMP molecular pathway overlaps to some extent with the MSI pathway because of the presence of sporadic MSI-H CRCs.

## 5. Molecular Features of Serrated Colorectal Lesions

### 5.1. Hyperplastic Polyps

The abovementioned WHO 2019 histopathological classification [11] distinguishes two types of HPs: GCHPs and MVHPs (Table 2). Identification of specific molecular biomarkers can facilitate the differential diagnosis between SSLs and HPs and, furthermore, between HP subtypes, which is challenging because of similar endoscopic features. At the molecular level, an MVHP usually bears a BRAF V600E mutation and CIMP-H without promoter hypermethylation of the MLH1 gene and without microsatellite instability; for that reason, it is considered a precursor of SSLs [81]. In both MVHPs and SSLs a hypomethylation of the MUC5AC gene, which regulates expression of the gastric-like mucin 5AC, can be found and seems to occur early in the serrated pathway. This finding highlights the possibility that MUC5AC could be a potential marker to evaluate the carcinogenetic risk [4]. Conversely, GCHPs often show K-ras and CIMP-L mutations, but not hypermethylation of MLH1 and MMS [82]. Furthermore, mutations in BRAF or K-ras are mutually exclusive in all serrated lesions and constitutively activate the MAPK-ERK pathway, inducing apoptosis arrest, colonocyte proliferation and p16 and IGFBP7 over-expression [81].

### 5.2. Sessile Serrated Lesions

At the molecular level, SSLs, whose precursor has been identified in MVHPs, exhibit BRAF V600E mutation and CIMP-H, microsatellite stability and unmethylated MLH1 [83]. BRAF mutation, along with CIMP-H, is regarded as a molecular hallmark of the “sessile serrated pathway”, leading from normal mucosa to MVHP, then to SSLs and subsequently to CRC (Figure 6) [84]. BRAF mutation and CIMP-H are tightly associated in SSLs and, even if they probably synergically facilitate carcinogenesis, the underlying mechanism remains unclear. In a recent paper that attracted attention, Tao et al. [85] found that in mice, aging-related spontaneous hypermethylation and consequent silencing of specific genes could activate the APC-WNT-pathway, causing differentiation defects. These changes make cells more sensitive than young ones to BRAF-induced carcinogenesis, producing adenocarcinomas with extensive abnormal gene-promoter CpG-island methylation, or methylator phenotype (CIMP) [85]. This suggests that a BRAF mutation is not a prerequisite for CIMP development in SSLs, where it is found in 63% to 100% [86,87]. Many previous studies have tried to define the molecular and protein expression profiles of SSLs. As previously pointed out for MVHPs, hypomethylation of the mucin 5AC gene (MUC5AC) with aberrant expression can be found in SSLs. Caruso et al. [88] found upregulation of cathepsin E (CTSE) and trefoil factor 1 in SSLs. In other studies, overexpression of trefoil factor 2 (TFF2) and v-set and immunoglobulin domain containing 1 (VSIG1) were described [56,89]. Even annexin A10 (ANXA10) has been proposed as a potential marker of SSLs. Recently, a new and more specific potential biomarker [90], Agrin (AGRN), has been identified, whose expression in the muscularis mucosa seems to be a specific feature of SSLs [90]. Hes-1, a target of the Notch pathway, is down-expressed in SSLs, when compared with normal tissues and HPs [91]. LOH or promoter hypermethylation of SLIT2 has been identified as another molecular marker of SSLs [92].

### 5.3. Sessile Serrated Lesions with Dysplasia (SSLDs)

The development of CRCs through SSLs implies the acquisition of a dysplastic pattern leading to SSLDs. At the molecular level, SSLDs exhibit BRAF V600E mutation, which, along with CIMP-H, is regarded as a molecular hallmark of the “sessile serrated pathway” [93]. CIMP-H status causes silencing of many cancer-related genes, by hypermethylation of their promoter regions. In order to obtain a dysplastic pattern, MLH1 gene silencing by the abovementioned mechanism seems to be needed [94]. MLH1 silencing causes MSI-H status with genome-wide microsatellite sequence alterations and mutations in cancer-related genes [93]. The high oncogenic pressure due to the acquisition of MLH1 silencing and MSI status in SSLs represents a molecular hallmark of progression to SSLDs which can rapidly evolve into CRCs. A plethora of studies have tried to assess the molecular status of SSLDs. Interestingly, MLH1 hypermethylation and CIMP-H pattern seem to occur in proximal colon large SSLs in elderly patients [95,96]. Lee et al. [95] analyzed 132 SSLs with no dysplastic pattern and, in order to define their high risk of evolving into SSLDs, tested these SSLs for MLH1 hypermethylation and CIMP-H. SSLs with these characteristics were found exclusively in patients >50 years, in the proximal colon (cecum, ascending colon, or transverse colon), whose histologically measured sizes were >5 mm (100%). Recent evidence found that many non-dysplastic SSLs can show, in few non-dysplastic crypts, partial loss of MLH1 expression and lower levels of MLH1 promoter methylation [95]. This is supposed to be a sign of forthcoming dysplastic evolution of SSLs. Because of MSI-H status, SSLDs harbor many other different genetic mutations [94]. Among others, a high mutational rate of FBXW7 and alterations in WNT-pathway-associated genes (RNF43, APC, AXIN2, MCC) have been reported in SSLDs with MLH-1 silencing and MSI-H [95,97,98,99]. The serrated pathway has two end results that differ in their clinical and prognostic features as well as in their molecular profile: the serrated adenocarcinoma (SAC) or the sporadic colorectal carcinoma showing molecular features of MSI-H.

### 5.4. Traditional Serrated Adenomas (TSAs)

The understanding of TSA molecular features and their progression to CRC, which have for a long time been enigmatic, has made remarkable progress (Figure 6). The carcinogenetic pathways that drive malignant transformation of normal colorectal mucosa to TSA, and then to carcinoma, was called “traditional serrated pathway” [100]. Nowadays, TSA molecular pathogenesis is supposed to proceed according to two different pathways: the K-ras mutation pathway and the BRAF mutation pathway [28,101]. When a normal mucosal cell acquires a K-ras mutation, it can convert into a K-ras-mutated TSA. TSAs bearing K-ras mutations are usually located in the distal colon [16]. Since TSAs are most frequently found in the left colon, the majority of TSAs follows the K-ras-mutation pathway. K-ras-mutated TSAs are generally CIMP-L or CIMP, and MSS [28,101]. On the contrary, normal colon mucosa cells with BRAF mutation can evolve into BRAF-mutated TSAs, which are more likely located in the proximal colon [102]. Much evidence supports the theory according to which BRAF-mutated TSAs derive from SSLs or HPs. Indeed, even SSLs and HPs, as previously mentioned, are more likely located in the proximal colon. Moreover, SSLs and HPs can be typically found close to BRAF-mutated TSAs [28,97] and BRAF-mutated TSAs frequently show CIMP-H just like SSLs and HPs. The main difference between SSLDs and BRAF-mutated TSAs is that in the latter MLH1 expression is preserved, notwithstanding the CIMP-H pattern [28]. In both cases (BRAF and K-ras-mutated TSAs) the transition of TSAs toward dysplasia is characterized by nuclear β-catenin accumulation, and WNT-signaling pathway impairment [28,101]. Unlike conventional adenomas, WNT-pathway activation is not due to APC inactivation but is more frequently due to PTPRK-RSPO3 fusions or RNF43 mutations [103]. Moreover, these genetic alterations were found almost exclusively in TSA and were mutually exclusive [103]. Subsequently it was demonstrated that PTPRK-RSPO3-fusion-positive TSAs were more frequently larger, with distal location and KRAS-mutated, whereas RNF43 mutations were more likely found in BRAF-mutated TSAs [104,105]. Like SSLs, TSAs aberrantly express gastric proteins including MUC5AC, ANXA10 and TFF2, but their positivity is significantly lower than that of HPs and SSLs. No specific biomarkers for TSAs have been established. In a recent study, Sohier et al. [106] found that LEFTY1, a protein down-regulating the TGF-β pathway, was overexpressed specifically in TSAs. This may play a promising role in the molecular diagnosis of TSAs. Both BRAF and K-ras-mutated TSAs can progress to carcinoma by TP53-inactivating mutations [62].

### 5.5. Serrated Polyposis Syndrome (SPS)

The genetic background of SPS remains unknown. Since the phenotype of serrated lesions in SPS is extremely variable and very little information is known about the underlying genetic mechanisms, SPS is probably the common result of more than one disorder. A small proportion of patients (<3%) affected by SPS display a germline mutation in RNF43, involved in the WNT pathway. However, most cases of SPS are not associated with any specific genetic variants [107]. Kokko et al. reported a germline mutation in EPHB2 as potentially causative in an individual with SPS [108]. Other genes with a potential role in the pathogenesis of SPS include ATM, PIF1, RBL1, TELO2 and XAF1 [107]. Nonetheless, these results should be confirmed in other studies and their role in carcinogenesis remains unconfirmed.

## 6. CRC Molecular Subtypes and the Role of Colorectal Serrated Lesions as Precursor

In 2015, Phipps et al. [109,110] proposed a CRC classification of five molecular subtypes on the bases of their MSI, CIMP and K-ras/BRAF mutation profiles. The five molecular subtypes were defined as follows [108,109]: 

*Type 1:* MSI+, CIMP+, BRAF-mutated, K-ras-wildtype. Type 1 CRC indicates sporadic MSI+ (MSI-high) tumors caused by MLH1 inactivation due to its promoter hypermethylation. This specific molecular pattern (MLH1 sporadic methylation, CIMP+) along with BRAF V600E mutation occurs almost exclusively in SSLs. Thus, SSLs can be considered as precursors of type 1 CRCs [109,110].

*Type 2:* MSI−, CIMP+, BRAF-mutated, K-ras-wildtype. This molecular subtype of CRCs with microsatellite stability, despite CIMP+ status and BRAF-activating mutation, matches the molecular background of BRAF-mutated TSAs and SSLs without dysplasia (which requires MLH1-methylation). Therefore, type 2 CRCs find their precursors in these two types of serrated lesions [109,110].

*Type 3:* MSI−, CIMP−, BRAF-wildtype, K-ras-mutated. This CRC subtype bears K-ras mutations, and their major precursors are represented by distally located KRAS-mutated TSAs. K-ras mutations can be found in a minor group of conventional adenomas, which can represent a premalignant lesion of this subgroup of CRC [109,110].

*Type 4:* MSI−, CIMP−, BRAF-wildtype, K-ras-wildtype. Type 4 CRCs represent the most common subtype which develops through the adenoma–carcinoma morphologic sequence. The first molecular step for carcinogenesis in this subtype is chromosomal instability, and progression to CRCs includes the development of a conventional adenoma (tubular, villous or tubulovillous) [109,110].

*Type 5:* MSI+, CIMP−, BRAF-wildtype, K-ras-wildtype. Most of these tumors arise in patients affected with Lynch syndrome, whose MSI-H status is not due to promoter hypermethylation of MMR genes, but to germline mutations in one of the MMR genes. Lynch-syndrome-associated CRCs usually develop through the conventional adenoma–carcinoma pathway. Thus, precursors are histologically conventional-type adenomas [109,110].

This classification turned out to be clinically relevant, because subtypes of CRCs were demonstrated to have a prognostic relevance [109,110]. Type 2 CRC was identified as the molecular subtype with the worst prognosis, whereas type 1 showed the best prognosis. Sorted by their prognostic characteristics, CRC subtypes was ranked (best to worst) as follows: type 1–type 5–type 4–type 3–type 2 [109,110]. Since a molecular MSI-H pattern is defined as a favorable prognostic factor in CRC [94] both type 1 and type 5 CRCs generally show a better prognosis then other subtypes [11]. According to this classification, SSLs are confirmed to be a heterogeneous group even on the prognostic level, since they represent the main precursors of either the best or the worst prognostic CRCs (type 1 and type 2). Type 1 and type 2 CRCs share a CIMP+/BRAF-mutated status, but the main difference between the two groups is the presence or absence of MSI, which mainly influences their prognostic value [111]. An SSL that develops MLH1 inactivation is at a high risk of progressing into an advanced lesion and then into a CRC; nonetheless, this subtype of CRC is supposed to have a favorable prognosis. Conversely, CIMP+/BRAF mutated/MSI- SSLs are considered precursors of poor-prognostic type 2 CRC. Thus, in order to avoid the development of poor-prognostic CRCs, it can be useful to screen CIMP+ SSLs without MLH1 methylation among nondysplastic SSL by molecular and immunohistochemical analyses [112]. 

## 7. Discussion and Conclusions

The majority of CRCs develop from benign adenomas as well as colorectal polyps [2]. Histologically, colorectal polyps are divided into adenomas, colorectal serrated lesions and other rare polyps, such as juvenile and inflammatory ones. In this context, carcinogenic risk is extremely variable even if adenomatous polyps and sessile serrated lesions are the most likely precursors of CRCs.

Until 2010, colorectal serrated lesions were generally considered as harmless lesions and reported as hyperplastic polyps (HPs) [9]. Thereafter, a multitude of terms were coined to describe these lesions, thus inducing strong misunderstandings about their terminology and classification. On these bases, a rational classification, from a histopathological point of view, was provided in 2010 by the World Health Organization (WHO) [10]. Colorectal serrated lesions should nowadays be classified according to the WHO classification of tumors of the digestive system, which was updated in 2019 [11]. The major edit in terminology is the approval of the new term “sessile serrated lesions” (SSLs), which aims to overcome the previous confounding terminologies “sessile serrated adenoma/sessile serrated polyp”. Indeed, the term “adenoma” encompasses the concept of dysplasia, which is not shown in a large number of SSLs. The term “lesion”, moreover, replaces the plethora of terms used in the past and appears both biologically and pathologically accurate [12]. Finally, the finding of a dysplastic pattern in SSLs requires the specific diagnostic terminology of “SSL with dysplasia”.

These recent advances in this field have induced a challenging impact on healthcare for gastroenterologists, pathologists and basic scientists, since, whilst a lot of studies have clearly described the mechanisms of the adenoma–carcinoma sequence, less evidence has been retrieved about serrated-polyp-related CRC. Unlike CRCs arising through the adenoma–carcinoma pathway, APC-inactivating mutations are rarely found in the serrated neoplasia pathway. Moreover, when APC mutations are found, they do not act as a first step to carcinogenesis. Most CRCs arising through the serrated neoplasm pathway carry a BRAF mutation, whereas K-ras mutations, the main step of the adenoma–carcinoma sequence, are markedly less frequent [16]. Furthermore, mutations in BRAF and K-ras are mutually exclusive in all serrated lesions. 

CRCs arising from serrated lesions originate via two different molecular routes, namely sporadic MSI and CpG island methylator phenotype (CIMP), which is considered as the major mechanism. CIMP involvement may be high or low (CIMP-H and CIMP-L). These main pathways (BRAF mutation, MSI and CIMP) are differently expressed in the four categories of serrated colonic lesions foreseen by the 2019 classification in the progression to cancer (“the serrated pathway”; Figure 6). Moreover, other molecular mechanisms have been described in serrated-lesion-related CRC, although they need to be confirmed in other studies and their role in carcinogenesis remains unconfirmed.

Finally, in 2015, Phipps et al. [109,110] proposed a CRC classification of five molecular subtypes on the bases of their MSI, CIMP and K-ras /BRAF mutation profiles and their prognostic characteristics. According to this classification, SSLs are confirmed to be a heterogeneous group even on the prognostic level, since they represent the main precursors of both the best and the worst prognosis.

In conclusion, serrated colorectal lesions, once defined as hyperplastic polyps with little or no neoplastic aptitude, nowadays constitute a variegated set of lesions susceptible to neoplastic transformation with sometimes favorable and sometimes very unfavorable prognoses. Therefore, the knowledge of the molecular pathways of their neoplastic evolution is an important research field not only for basic science, but also for possible clinical implications, on the basis of which precise follow-up protocols of screening should be defined.

## Figures and Tables

**Figure 1 ijms-23-04461-f001:**
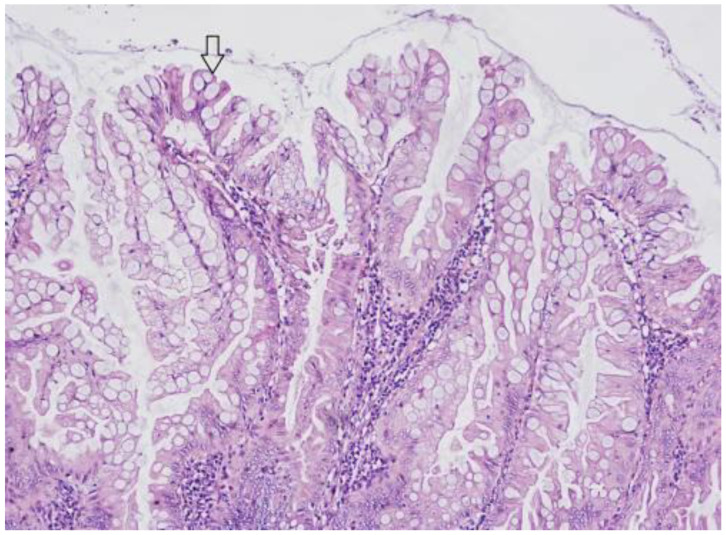
Histopathological picture of a hyperplastic polyp. The arrow highlights mucin vescicles. Hematoxylin eosin stain, magnification 200×.

**Figure 2 ijms-23-04461-f002:**
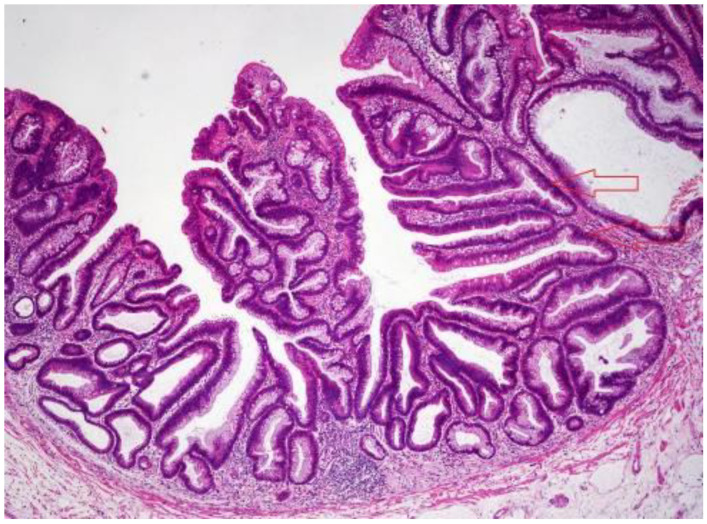
Histopathological picture of an SSL. L- and T-shaped crypts are highlighted by arrows. Hematoxylin eosin stain, magnification 40×.

**Figure 3 ijms-23-04461-f003:**
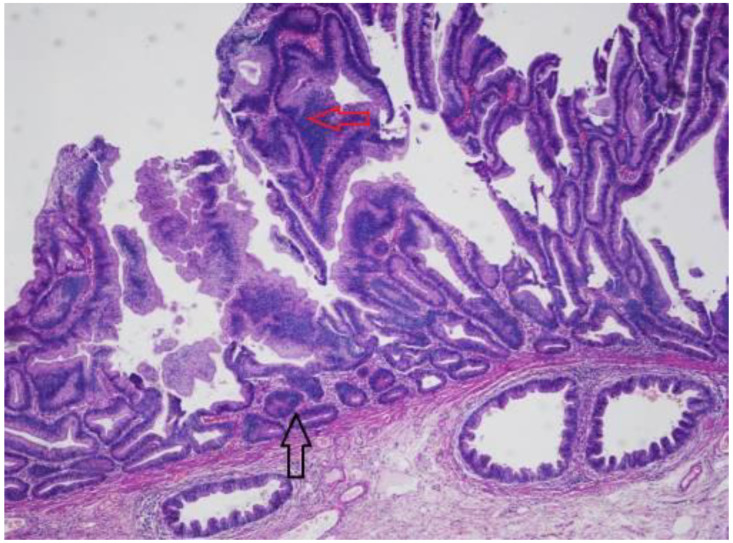
Histopathological picture of an SSL with dysplasia (red arrow) and pseudo-invasive pattern (black arrow). Hematoxylin eosin stain, magnification 40×.

**Figure 4 ijms-23-04461-f004:**
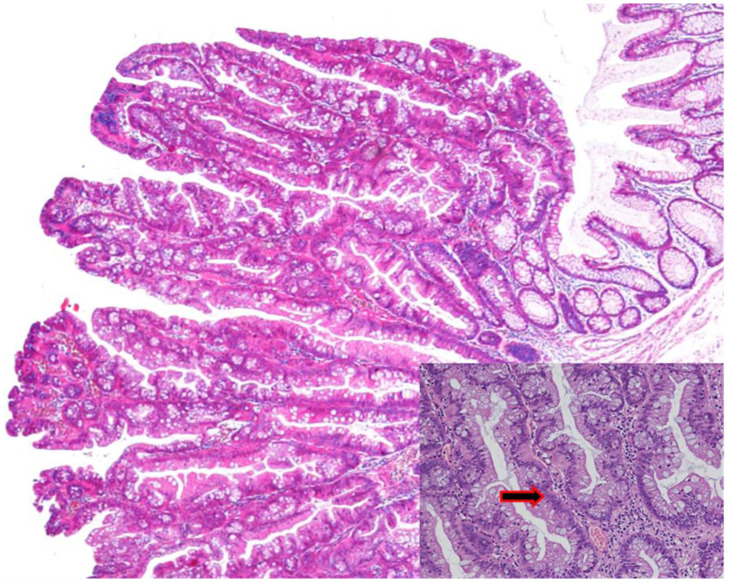
Histopathological picture of a TSA. Penicillate nuclei are indicated in the insert by an arrow. Hematoxylin eosin stain, magnification 20×.

**Figure 5 ijms-23-04461-f005:**
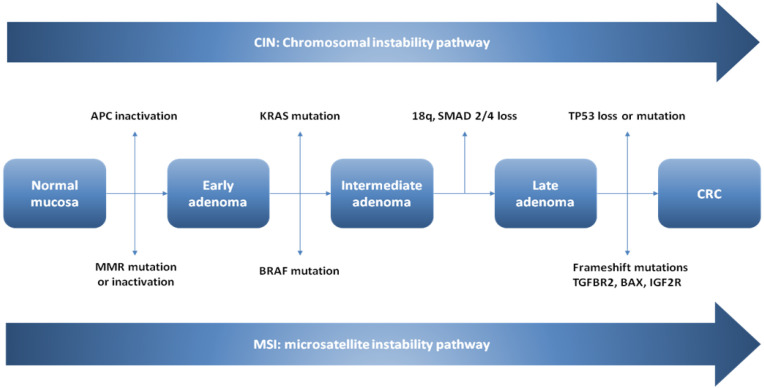
Adenoma–carcinoma sequence: molecular and morphologic pathways.

**Figure 6 ijms-23-04461-f006:**
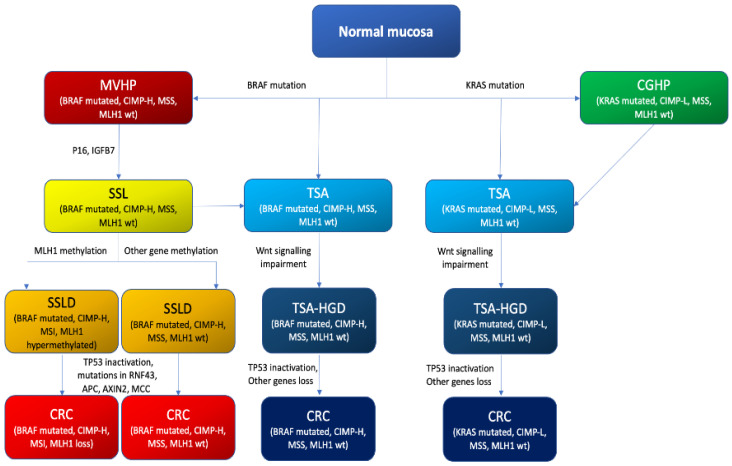
Sessile serrated lesion and traditional serrated adenoma pathogenesis: the “serrated pathway”.

**Table 1 ijms-23-04461-t001:** Classification of serrated colorectal lesions (4th edition).

Serrated Colorectal Lesions Classification (2010 WHO 4th Edition)	
**Histological type**	**Histological sub-type**
Hyperplastic polyp (HP)	Microvescicular type (MVHP) Goblet-cell rich type (GCHP) Mucin-poor type (MPHP)
Sessile serrated adenoma/polyp (SSA/P)	SSA/P with dysplasia SSA/P without dysplasia
Traditional serrated adenoma (TSA)	

**Table 2 ijms-23-04461-t002:** Classification of serrated colorectal lesions (5th edition). Text in italics identifies novel categories.

Serrated Colorectal Lesions Classification (2019 WHO 5th Edition)
**Histological type**	**Histological subtype**
Hyperplastic polyp (HP)	Microvescicular type (MVHP)Goblet-cell rich type (GCHP)
Sessile serrated lesion (SSL)	SSLSSL with dysplasia (SSLD)
Traditional serrated adenoma (TSA)	
Serrated adenoma, unclassified

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
