# Peer review of "Serrated Colorectal Lesions: An Up-to-Date Review from Histological Pattern to Molecular Pathogenesis"

_ijms, 2022, doi:10.3390/ijms23084461_

Round 1
Reviewer 1 Report
A review article: Serrated colorectal lesions: an up-to-date review from histological pattern to molecular pathogenesis, (written by Mezzapesa et al.) belongs to topic subjects and clearly demonstrates its need in categorizing of serrated pre-cancerous lesions to colorectal cancer. I like to way how authors carry readers from past to present with WHO classification terms, it is very educational for the serrated colorectal cancer field.
Figures in the article are excellent, but I would see a need in editing of tables 1 and 2, so they become clearer and easier to read and understand. Throughout the text there are several words without a space bar, so editing of that is also needed. In Chapter 5.5 Serrated polyposis syndrome (SPS), in 5th row, there is confusion in sentence "...WNT pathway and this is the only However, most cases of SPS...", which should be correct.
In Chapter 4.1 Hereditary CRCs, I am sure, that authors are aware that at least couple of other hereditary CRC syndromes exist than just FAP, Lynch and hamartomatous polyposis syndrome, like Juvenile Polyposis Syndrome and Polymerase Proofreading associated Polyposis. I feel that these syndromes (at least) need to mention, even though it does not have a mean of the topic itself.
Author Response
Questions and point by point answers are attached

Reviewer 2 Report
I wish to thank and congratulate the authors for this very comprehensive review of the different types of lesions found in the colorectal tissues, especially when it comes to the serrated CR lesions. It is very interesting to read about the differential molecular aspects as well.
I only have a few suggestions for further improving this review, in order to make it even more reader-firendly:
1) Can you add arrows and notes on the Figures 1-4, to point out the main morphological features?
2) On Figure 5, please indicate also the type of genomic changes for the 3 genes (TGFBR2 etc.).
3) On Figure 6, going horizontally, the molecular changes are/seem identical for the different types of lesions. Without making the Figure too busy, would it be possible to add some of the other mutations etc found in the various lesion types?
4) Even better, and as additional info I was missing in Figure 6, I think the readers would appreciate a table with all typical molecular aberrations and also the locations of the type of lesions where they are typically found.
Author Response
Questions and point by point answers are attached.
